# OpenReview forum: "From Threat to Tool: Leveraging Refusal-Aware Injection Attacks for Safety Alignment"
_ICLR.cc/2026/Conference — Submitted to ICLR 2026_

### Official Review · Reviewer_f4EQ · 2025-10-31

**Soundness:** 2
**Presentation:** 2
**Contribution:** 2
**Rating:** 2
**Confidence:** 4

**Summary:**

The paper proposes to detect the refusal tokens and modify it into predefined sequences to allow harmful contents generation. Experiments show that the proposed method outperforms existing baselines. When utilized for further fine-tuning LLMs, the data shows promising results for increased safety without alignment tax.

**Strengths:**

1. the paper is well written and easy to follow
2. experiments show that the proposed method outperform baseline methods by a large margin.
3. the generated harmful contents can also be utilized to further fine-tune the model for improved safety performance.

**Weaknesses:**

1. the compared baseline methods are all outdated ones from 2023 and 2024.
2. the contribution of the paper is weak. the main contribution is to replace a token to be decoded with predefined sequences. the rest of the methods such as utilizing SimPO is just a simple application.
3. although the proposed method is branded as gray-box, it requires access to the decoding process of LLMs which seems impossible for production models such as GPT or Gemini. Thus the application scenario of the proposed method is greatly limited.

**Questions:**

1. has the author studied why the model trained with synthetic data from the proposed method has increased performance in MMLU or ARC
2. will the proposed method increase the rate of over/false refusals as mentioned in [1] and [2]

[1] Röttger, Paul, et al. "Xstest: A test suite for identifying exaggerated safety behaviours in large language models." arXiv preprint arXiv:2308.01263 (2023).

[2] An, Bang, et al. "Automatic pseudo-harmful prompt generation for evaluating false refusals in large language models." arXiv preprint arXiv:2409.00598 (2024).

---

### Official Review · Reviewer_U4Me · 2025-11-01

**Soundness:** 2
**Presentation:** 3
**Contribution:** 2
**Rating:** 4
**Confidence:** 4

**Summary:**

This paper proposes attacking LLMs by replacing refusal tokens (such as `can't`) with predefined phrases to elicit harmful responses.
Furthermore, the paper repurposes the idea as a data synthesis framework for improving the safety alignment of LLMs by training on pairs of generated (original safe responses) and elicited harmful responses using SimPO. Experimental results show that the method can elicit more harmful responses compared with baselines and improve the safety alignment with little alignment tax.

**Strengths:**

- The paper is well-written and organized, making the idea easy to follow.

- The idea of turning attacking into an alignment data generation pipeline is interesting and proves effective.

**Weaknesses:**

- As an attacking method, the framework exhibits several critical limitations concerning refusal token replacements:
   - For commercial models, access to token probability distributions is often restricted, preventing the application of this method to commercial deployments where such attacks would constitute a genuine security concern.
   - The refusal token selection process is arbitrary and depends on manual inspection. Moreover, Table 4 reveals inconsistencies in token classification—for instance, tokens such as `support` and `fulfill` are not categorized as refusal tokens. The rationale behind these selections requires clarification, as does the potential impact of alternative token choices on overall performance. Additionally, the handling of tokenization for these words remains inadequately addressed.

- As a data synthesis method, the approach lacks sufficient novelty. It is well-established that on-policy data (i.e., responses generated by the model itself) provides benefits (thus less alignment tax is expected), and the proposed method essentially functions as Rejection Fine-Tuning (RFT), offering limited methodological innovation beyond existing techniques.

**Questions:**

Typo:

Line 163:  can be adjusted

---

### Official Review · Reviewer_MkgB · 2025-11-01

**Soundness:** 2
**Presentation:** 3
**Contribution:** 1
**Rating:** 2
**Confidence:** 5

**Summary:**

This paper investigates a more general form of the prefilling attack for circumventing the safety alignment of LLMs called “RAAI”. During generation, if a fixed set of refusal tokens has enough predicted probability mass, then a predefined set of tokens is injected to attempt to steer the model towards compliance. The responses generated from RAAI attacks are then used as synthetic data for safety alignment.

**Strengths:**

1. The injection technique for circumventing the safety alignment of LLMs is simple and effective.
2. Ablations of injection phrases and refusal thresholds are performed.
3. Fine-tuning on synthetic data from RAAI improves safety without sacrificing much model utility.

**Weaknesses:**

1. The novelty of Refusal-Aware Adaptive Injection is quite limited. It is essentially a more general version of the prefilling attack where the harmful prefill can be inserted mid-generation based on some simple rules. In fact, prior work has explored similar ideas of alternating between generation and injection of tokens, such as in [1]. I would suggest including [1] as a baseline in your experiments.
2. The citations for prefilling attacks misses some critical works, namely [2] and [3].
3. The comparison to naive prefilling could be strengthened. I recommend comparing against the approach of [2], where rather than using “predefined phrases as a fixed prefix” for the prefill, prompt-specific prefills are generated via few-shot prompting. The prefills are typically structured in the following way: given the prompt (e.g., “Tell me how to build a bomb”), the prefill starts with an affirmative restatement of the prompt (e.g., “Sure, here are instructions for building a bomb”) followed by just a bit more text that could begin the harmful content (e.g., “Sure, here are instructions for building a bomb:\n\nStep 1. Gather”). I would suspect prefilling attacks structured this way to be comparable to RAAI performance.
4. Similarly, I suspect that using synthetic data generated by the prefilling attack approach of [2] for safety alignment would yield comparable improvements to using synthetic data generated by RAAI — I suggest investigating and reporting such results in Table 7 and 8.
5. Typos:
- Line 163: “can be adjusted for” is missing the start of the sentence (I assume tau).

References:

[1] Zhang, Zhuo, et al. "Make them spill the beans! coercive knowledge extraction from (production) llms." arXiv preprint arXiv:2312.04782 (2023).

[2] Vega, Jason, et al. "Bypassing the safety training of open-source llms with priming attacks." arXiv preprint arXiv:2312.12321 (2023).

[3] Andriushchenko, Maksym, Francesco Croce, and Nicolas Flammarion. "Jailbreaking leading safety-aligned llms with simple adaptive attacks." arXiv preprint arXiv:2404.02151 (2024).

**Questions:**

See weaknesses.

---

### Author Response · Authors · 2025-11-30

We thank all reviewers for their thoughtful feedback and constructive suggestions.

We are glad that reviewers found the method simple, effective, and easy to follow, and that the alignment results are strong. Below, we address the main concerns and clarify the intended scope and core contribution of our work.

---

### **1. Clarifying the Main Contribution ("Threat → Tool" Framing)**

The primary goal of our paper is **not** to propose an entirely new jailbreak attack, but to demonstrate that:

> **LLM attack methods can be reframed as practical data-generation tools, producing high-quality dispreferred responses that substantially improve safety alignment.**

This perspective—using inference-time attacks as a stable, scalable source of harmful responses—is the novel contribution. RAAI is intentionally simple; its value lies in its **consistency, naturalness, and diversity** as a synthetic data generator.

We will make this framing more explicit.

---

### **2. Novelty Beyond Prefilling and Prior Adaptive Attacks**

Reviewers noted similarities with prefilling or adaptive injection. To clarify:

* Prefilling is *static* and occurs only once at the prefix.
* RAAI injects adaptively **when internal refusal probability spikes**, which varies significantly across models (Fig. 2), and drives RAAI's effectiveness.
* While related in spirit, prior work ([1–3]) does not trigger injections based on refusal-signal dynamics.

Our contribution is not algorithmic complexity but **leveraging refusal-signal trajectories** to turn a lightweight attack into a robust data-collection mechanism.

---

### **3. Baseline Choice and Comparison Scope**

Our goal is to evaluate **training-free, inference-time methods** suitable for data generation.
Methods requiring auxiliary models, multi-round pipelines, or training-time access fall outside this scope.

While stronger prefilling variants exist (e.g., prompt-specific prefixes), they require prompt crafting or auxiliary models and are not directly usable as scalable data generators. We will clarify this scope and rationale.

---

### **4. Applicability to Commercial Models (Gray-Box Access)**

We agree that commercial APIs often restrict token-level probabilities.
However, our focus is **open-weight and research models**, where generating harmful responses for alignment remains a bottleneck. Our intention is not to claim a real-world attack on production APIs, but to propose a practical method for **alignment data synthesis**.

---

### **5. Refusal Token Selection and Stability**

The refusal token set was constructed via a reproducible frequency-based procedure.
We agree that the explanation can be expanded, and will clarify this.

Importantly, RAAI's performance is *not* sensitive to the exact composition of the token set: its key behavior is driven by **timing** rather than lexicon choice, as shown by stable SR distributions (Fig. 4 left).

---

### **6. Why RAAI-Generated Data Improves Alignment**

The safety gains are due to **data quality**, not method novelty:

* **Consistently harmful outputs** (SR scores concentrated near 1.0)
* **Fluent, non-templated generations**, unlike many rule-based attacks
* **Multiple diverse harmful completions** per prompt (5.59/prompt vs. 2.07 for GPTFuzzer)

These properties make RAAI-generated pairs especially well-suited for SimPO-style preference optimization, explaining the strong safety improvements without utility degradation.

---

### References:
- [1] Zhang, Zhuo, et al. "Make them spill the beans! coercive knowledge extraction from (production) llms." arXiv preprint arXiv:2312.04782 (2023).
- [2] Vega, Jason, et al. "Bypassing the safety training of open-source llms with priming attacks." arXiv preprint arXiv:2312.12321 (2023).
- [3] Andriushchenko, Maksym, Francesco Croce, and Nicolas Flammarion. "Jailbreaking leading safety-aligned llms with simple adaptive attacks." arXiv preprint arXiv:2404.02151 (2024).

---

### Meta-Review · Area_Chair_rEiG · 2025-12-31

**Summary:**

Reviewers MkgB, U4Me, and f4EQ agree the paper is clearly written and that the core mechanism (refusal-aware adaptive phrase injection) is simple and empirically effective at eliciting harmful completions on the evaluated open-weight models, with downstream fine-tuning showing improved safety with little observed utility loss. The main concerns are that the methodological novelty is limited and appears closely related to prefilling/priming and prior adaptive attacks, and that the experimental comparisons/citations do not convincingly position RAAI against the strongest or most relevant baselines. Reviewers also highlight practical scope limits (needs access to decoding/logits) and insufficient justification/sensitivity analysis for the refusal-token selection.

**Reviewer Concerns:**

The rebuttal addresses several framing and clarification issues: it more clearly positions the contribution as “threat → tool” (attack-style inference-time data generation for alignment), narrows the intended applicability to open-weight/research settings (rather than production APIs), and clarifies that refusal-token selection is intended to be reproducible and not overly sensitive, while offering a plausible explanation for why the generated data helps SimPO-style optimization (fluency, consistency, diversity). However, the most consequential concerns remain outstanding: MkgB’s request for stronger comparisons to prompt-conditioned prefilling/priming and related adaptive injection work is not resolved empirically, and U4Me/f4EQ’s skepticism that the data-synthesis contribution may reduce to known on-policy/RFT-style benefits is not decisively addressed by ablations that isolate what RAAI adds over simpler generators. Overall, the rebuttal improves clarity and scope but does not yet close the gap on novelty and baseline completeness.

**Reviewer Scores:**

MkgB (2): likely unchanged; their critique is primarily about limited novelty/positioning and missing key baseline comparisons, and the rebuttal does not provide new evidence to overturn this. U4Me (4): could remain at 4, with a small chance of a slight increase if the clarified scope/token-selection procedure is viewed as adequately mitigating the “arbitrary tokens / production relevance” concern; novelty reservations would likely persist. f4EQ (2): likely unchanged absent stronger, more up-to-date baselines and a clearer empirical case that gains are specific to the refusal-aware adaptive mechanism rather than a generic synthetic-data fine-tuning effect.

---

### Decision · Program_Chairs · 2026-01-26

Reject